# Lysophosphatidylserine Induces MUC5AC Production via the Feedforward Regulation of the TACE-EGFR-ERK Pathway in Airway Epithelial Cells in a Receptor-Independent Manner

**DOI:** 10.3390/ijms23073866

**Published:** 2022-03-31

**Authors:** Myeong Seong Sim, Hye Jeong Kim, Sang Hee Jo, Chun Kim, Il Yup Chung

**Affiliations:** 1Department of Bionano Technology, Hanyang University, Ansan 15588, Korea; shimms44@gmail.com (M.S.S.); khjeong1233@gmail.com (H.J.K.); 2Department of Molecular and Life Sciences, College of Science and Convergence Technology, Hanyang University, Ansan 15588, Korea; sanghee0496@naver.com

**Keywords:** airway epithelial cells, EGFR, MUC5AC, LysoPS, TACE, TGF-α, TLR2

## Abstract

Lysophosphatidylserine (LysoPS) is an amphipathic lysophospholipid that mediates a broad spectrum of inflammatory responses through a poorly characterized mechanism. Because LysoPS levels can rise in a variety of pathological conditions, we sought to investigate LysoPS’s potential role in airway epithelial cells that actively participate in lung homeostasis. Here, we report a previously unappreciated function of LysoPS in production of a mucin component, MUC5AC, in the airway epithelial cells. LysoPS stimulated lung epithelial cells to produce MUC5AC via signaling pathways involving TACE, EGFR, and ERK. Specifically, LysoPS- dependent biphasic activation of ERK resulted in TGF-α secretion and strong EGFR phosphorylation leading to MUC5AC production. Collectively, LysoPS induces the expression of MUC5AC via a feedback loop composed of proligand synthesis and its proteolysis by TACE and following autocrine EGFR activation. To our surprise, we were not able to find a role of GPCRs and TLR2, known LyoPS receptors in LysoPS-induced MUC5AC production in airway epithelial cells, suggesting a potential receptor-independent action of LysoPS during inflammation. This study provides new insight into the potential function and mechanism of LysoPS as an emerging lipid mediator in airway inflammation.

## 1. Introduction

Airway mucus is essential to protect a wide area of our respiratory system. Regulatory failure of airway mucus is often associated with a wide range of pathological respiratory abnormalities. During chronic inflammatory airway conditions, excessive mucus is poorly cleared, leading to mucus plugging, airway obstruction, and a decline in lung function and exacerbation [1]. Such mucus hypersecretion is one of the most important characteristics of allergic airway obstructive diseases, such as asthma, chronic obstructive pulmonary disease, and cystic fibrosis [2]. MUC5AC is one of the major gel-forming mucins that coats airway apical surfaces and is secreted by goblet cells within the bronchial epithelium [3,4]. Several lines of clinical and experimental studies demonstrate that MUC5AC is reportedly upregulated in asthmatics during allergic airway inflammation [5,6,7,8] and its overproduction is critical to the development of airway hyperresponsiveness in a murine model of asthma [7,8,9]. Furthermore, elevated levels of MUC5AC may contribute to the high viscosity of airway mucus and sputum retention in the airways of patients with coronavirus disease 2019 (COVID-19) [10]. Taken together, MUC5AC plays a central role in the pathology of airway obstructive diseases [11].

A variety of exogenous and endogenous insults provoke mucin hyperproduction in bronchial epithelium tissues during inflammatory reactions [12,13]. Diverse membrane receptors are known to sense these stimuli and activate signaling pathways leading to the production of MUC5AC [14,15,16,17,18]. Interestingly, some stimuli appear to instigate MUC5AC production without engaging their cognate receptors [19,20,21,22]. The multiple signals triggered by these stimuli can be convened to activate tumor necrosis factor-alpha converting enzyme (TACE), also known as disintegrin-like metalloprotease 17. TACE modulates the EGFR-dependent ERK signaling pathway by cleaving EGFR proligands such as transforming growth factor alpha (TGF-α) on airway epithelial cells [17,23]. The TACE-EGFR-ERK pathway is considered a major route for funneling signals from a variety of stimuli eliciting MUC5AC production [17,21].

Lysophosphatidylserine (LysoPS), a deacylated form of phosphatidylserine, is now recognized as an important class of a bioactive glycophospholipid mediator. Several GPCRs, including P2Y10, GPR34, GPR174, and G2A (also referred to as GPR132), have been identified as receptors of LysoPS [24,25]. In addition, TLR2 can be activated by LysoPS [26]. By taking advantage of the identified receptors, LyosPS can mediate a broad spectrum of biological responses [27]. Even though the level of LysoPS in human and mouse plasma can increase in certain pathological conditions [28,29], the mode of action and pathophysiolocal levels of LysoPS in in vivo circumstances are yet largely unknown. In this regard, the goal of this study was to look into a potential role of LysoPS during airway inflammation. Here, we present a new proinflammatory function of LysoPS as a potent inducer of MUC5AC production in airway epithelial cells. We also report that LysoPS triggers biphasic ERK phosphorylation through positive feedback control of the TACE-EGFR-ERK pathway in a receptor- and ROS-independent manner.

## 2. Results

### 2.1. LysoPS Induces MUC5AC Production in Airway Epithelial Cells 

One of the hallmarks of activation of lung epithelial cells is the overexpression of MUC5AC, a component of mucin. We examined whether LysoPS provoked MUC5AC production in an NCI-H292 human lung epithelial cell line. When NCI-H292 cells were stimulated with LysoPS (30 μM) and EGF (25 ng/mL), LysoPS induced expression of MUC5AC protein at a level comparable to that induced by EGF, which is considered the most potent and predominant stimulus of MUC5AC production [30,31], and at higher levels than those produced by ATP plus S100A9 (Figure 1A), which was shown to be capable of inducing MUC5AC production at a moderate level [32]. An ICC analysis revealed that the numbers of MUC5AC-positive cells in response to either LysoPS or EGF ranged from 20 to 40 per high-power field (HPF), while that of untreated cells was 0 to 5. The level of LysoPS-induced MUC5AC mRNA was significant at 4 h, reached a peak at 8 h, and declined thereafter (left panel in Figure 1B). LysoPS-induced MUC5AC mRNA was dose-dependent (right panel in Figure 1B), suggesting that LysoPS activates MUC5AC by transcriptional regulation. The viability of NCI-H292 cells was not compromised in response up to 30 μM LysoPS during an incubation period of 24 h, and declined to approximately 75% upon exposure to 50 μM LysoPS (Figure 1C). Because 30 μM LysoPS consistently induced strong MUC5AC production in NCI-H292 cells without a significant sign of cytotoxicity, this concentration was used for subsequent experiments, if not otherwise mentioned. To evaluate the capacity of LysoPS to induce MUC5AC expression in a more physiological setting, NHBE cells in an air-liquid interface culture were stimulated with LysoPS and the effects were examined. In keeping with the findings in NCI-H292 cells, LysoPS induced expression of MUC5AC mRNA and MUC5AC protein in NHBE cells (Figure 1D). Because some lysophospholipids can also serve as signaling molecules [28,33,34], we examined the effects of lysophosphatidylethanolamine (LysoPE) or lysophosphatidylcholine (LysoPC) on MUC5AC production. Despite the structural similarities of the three lysophospholipids, only LysoPS acted as a potent activator of expression of MUC5AC protein (Figure 1E).

### 2.2. LysoPS-Induced MUC5AC Production Is Blocked by Inhibitors of TACE, EGFR, and MEK1/2 but Not by Inhibitors of Caspase-1 or NF-κB

In response to myriad endogenous and exogenous stimuli, the ERK and NF-κB-dependent signaling pathways play central roles in the expression of MUC5AC [12,13,35]. Using pharmacological inhibitors, we examined the signaling pathways that could be responsible for LysoPS-induced MUC5AC production. Production of MUC5AC was almost completely inhibited by TAPI2 (a TACE inhibitor), AG1478 (an EGFR inhibitor), EGFR neutralizing antibody, U0126 (an MEK1/2 inhibitor) (Figure 2). Since TACE serves as the upstream regulator of EGFR signaling by shedding EGFR ligands [36], these results suggest that NCI-H292 cells may produce one of the EGFR ligands, and it is processed by the enzyme, TACE, in response to LysoPS. In contrast, SB203580, SP600125, Ac-YVAD-cmk, Z-VAD-fmk, and BAY11-7082, which inhibit p38 MAPK, JNK, caspase-1, pan-caspases, and NF-κB, respectively, did not prevent the expression of MUC5AC in response to LysoPS (Figure 2). As reported by previous studies, EGF-induced MUC5AC production was blocked by EGFR and MEK1/2 inhibitors and EGFR neutralizing antibody and the ATP and S100A9-dependent MUC5AC production was blunted by the NF-κB inhibitor (Figure 2) [15,18,32,36]. This result suggests that the TACE-EGFR-ERK axis is a key signaling pathway for LysoPS-induced MUC5AC production.

### 2.3. LysoPS Induces ERK Phosphorylation in a Biphasic Manner with Robust Activation of EGFR in the Late Phase

As the TACE-EGFR-ERK pathway appeared to be indispensable for LysoPS-induced MUC5AC production (Figure 2), we next examined whether LysoPS can activate this pathway. Stimulating NCI-H292 cells with LysoPS for 10 min induced ERK phosphorylation in a dose-dependent manner (Figure 3A) similar to what was seen in LysoPS-induced MUC5AC production (Figure 1E). Kinetic experiments showed that LysoPS induced ERK activation in a biphasic manner. The first phase of ERK phosphorylation occurred in 10 min following LysoPS stimulation and returned to baseline at 60 min. The second phase of ERK phosphorylation occurred moderately at 4–8 h following stimulation (Figure 3B). In contrast, EGF induced a single-phase of ERK phosphorylation (Figure 3C and data not shown). EGFR phosphorylation was marginal or absent in unstimulated cells (lane 1 in Figure 3B,C). Upon exposure to LysoPS for 10 min, EGFR phosphorylation was weak but evident (lane 2 in Figure 3B,C). Phosphorylation of EGFR was markedly increased 4 h after stimulation, reached its maximum level at 8 h, and lasted for 24 h (lanes 5–7 in Figure 3B). As expected, EGF induced a robust response in EGFR phosphorylation at 10 min. Next, pharmacologic inhibitors were used to confirm the involvement of TACE, EGFR, and ERK in LysoPS-mediated signaling. The early ERK phosphorylation (at 10 min) was inhibited substantially by TAPI2 (lane 3 in Figure 3C) and completely by AG1478 and U0126 (lanes 4 and 5 in Figure 3C), whereas the late ERK phosphorylation (at 8 h) was completely abolished by all three inhibitors (lanes 13–15 in Figure 3C). These results indicate that TACE, EGFR, and ERK are activated by LysoPS at both phases. The slight induction of EGFR phosphorylation at 10 min after LysoPS stimulation was blocked by the EGFR inhibitor but not the TACE inhibitor (lanes 1–4 in Figure 3C). The MEK inhibitor did not prevent EGFR phosphorylation (lane 5 in Figure 3C), which indicates that ERK is a downstream molecule of EGFR. In contrast to ERK phosphorylation, which was markedly inhibited by the TACE inhibitor, EGFR phosphorylation was not affected by the TACE inhibitor during the early phase upon exposure to LysoPS (lane 3 in Figure 3C). It is not clear why the TACE inhibitor failed to inhibit EGFR phosphorylation during the early phase, given the sequential activation of the TACE-EGFR-ERK pathway. In contrast, the EGFR phosphorylation at 8 h was partially but equally inhibited by both TACE and EGFR inhibitors (lanes 13 and 14 in Figure 3C). The concomitant appearance of moderate ERK phosphorylation and strong EGFR phosphorylation during the late phase suggests that TACE-dependent EGFR proligand cleavage could contribute to the late EGFR phosphorylation responses and subsequent MUC5AC production in NCI-H292 cells, as reported previously [17,21,22]. We also confirmed that LysoPS can induce EGFR phosphorylation in NHBE cells at a similar kinetics within NCI-H292 cells (Figure 3D).

### 2.4. Early-Phase ERK Phosphorylation Is at Least in Part the Result of Activation of EGFR

The results depicted in Figure 3C raise an intriguing question: Why was the robust ERK activation induced by LysoPS completely blocked by selective EGFR inhibitor AG1478 during the early phase (10 min), although LysoPS is conceivably unable to bind to and activate EGFR directly. Similar to the effect of AG1478, EGFR neutralizing antibody indeed completely inhibited ERK phosphorylation elicited by LysoPS in the early phase (Appendix A). Since protein kinase inhibitors such as AG1478 have a broad spectrum of selectivity, which is a consequence of the highly conserved ATP-binding sites shared by all protein kinases [37], we examined the phosphorylation of EGFR by LysoPS in the presence of varying concentrations of AG1478. The phosphorylation of EGFR was efficiently inhibited at all tested concentrations, suggesting that LysoPS activates ERK via EGFR in the early phase of stimulation (Appendix A). The results from these complementary experiments suggest that LysoPS induces ERK phosphorylation in the early phase, in part by EGFR activation. It remains unclear why weak EGFR phosphorylation is disproportionately colinear with the strong ERK phosphorylation in the early phase (lane 2 in Figure 3B).

### 2.5. The TACE-EGFR-ERK Pathway Is Required for MUC5AC Production during the Late Phase

Since the activation of ERK was abolished by inhibitors of TACE, EGFR, and MEK1/2 in the late phase (lanes 13–15 in Figure 3C), we examined whether the signaling axis is required for MUC5AC production. When cells were treated with individual inhibitors, even 1 or 3 h after LysoPS treatment, LysoPS-induced MUC5AC production was almost completely abolished by each inhibitor. However, after 8 h of LysoPS treatment, the production of MUC5AC in NCI-H292 cells was not efficiently prohibited by those inhibitors. The time-dependent effect of the inhibitors in MUC5AC production by LysoPS indicates that the late phase of ERK and EGFR activation determines the expression of MUC5AC in NCI-H292 cells (Figure 4). In order to ensure the contribution of TACE in LysoPS-induced signaling and MUC5AC production, we employed siRNA against TACE. When the expression level of TACE mRNA was significantly reduced by the specific siRNA treatment in NCI-H292 cells (Figure 5A), LysoPS-dependent activation of EGFR and the production of MUC5ACwas substantially decreased (Figure 5B,C). This result supports the finding that TACE plays a vital role in LysoPS-induced MUC5AC synthesis, as demonstrated by our experiment with a TACE-specific inhibitor (Figure 4).

### 2.6. LysoPS Induces TGF-α Production in a TACE-Dependent Manner

Since we observed the concurrent activation of ERK and EGFR in the late phase of LysoPS stimulation, we examined whether LysoPS induces known EGFR proligands and activates EFGR in an autocrine manner, as previously demonstrated [17,21,22]. In search of potential EGFR ligands produced by LysoPS, we found that LysoPS induced TGF-α, which is a potent inducer of mucin production in airway epithelial cells [17,20]. TGF-α was secreted at 30 min following treatment with LysoPS and increased steadily up to 24 h (Figure 6A). LysoPS-induced TGF-α production was dose-dependent (Figure 6B) and abolished by the TACE inhibitor (Figure 6C). Interestingly, the TGF-α neutralizing antibody only partially inhibited MUC5AC production (Figure 6D). At the moment, we cannot rule out the possibility that not just TGF-alpha, but also additional EGF family proteins may contribute to autocrine EGFR activation in LysoPS-triggered late-phase EGFR activation.

### 2.7. Neither GPCRs nor TLR2 Is Responsible for LysoPS-Induced ERK Phosphorylation and MUC5AC Production

To examine whether the activation of the TACE-EGFR-ERK pathway is mediated through known receptors of LysoPS, we investigated the expression of GPCRs, including P2Y10, GPR34, GPR174, and G2A [24,25]. Flow cytometry and western blot analyses revealed that none of these four receptors were expressed in NCH-H292 cells (Figure 7A and data not shown). Furthermore, the broad spectrum P2 purinergic receptor inhibitor suramin and more selective P2X receptor antagonist (A438079) did not inhibit LysoPS-induced MUC5AC production (Figure 7B), ruling out the possible involvement of ATP release and the subsequent activation of the receptors for mucin production. Since PTX inhibited LysoPS-induced ERK phosphorylation in GPR34-transfected cells [38], we also examined whether Gi/o activation is involved in the effects of LysoPS on airway epithelial cells. Both LysoPS-induced MUC5AC production and ERK activation were insensitive to PTX (Figure 7C). Lysophosphatidic acid (LPA) has been shown to induce ERK activation [39] and other cellular processes in a PTX-dependent manner [40]. Consistent with previous reports, LPA induced MUC5AC production and ERK phosphorylation in a PTX-dependent manner (Figure 7C), indicating that LysoPS induces MUC5AC production via a different signaling pathway from that utilized by LPA. Another potential LysoPS receptor TLR2 is expressed in airway epithelial cells [41] and participates in MUC5AC production through the transactivation of EGFR [16]. Although NCI-H292 cells express TLR2 (Figure 7D), a TLR2 antagonist C29 did not significantly inhibit LysoPS-induced MUC5AC production (Figure 7E). In addition, C29 did not inhibit LysoPS-dependent early- and late-phase of ERK phosphorylation and collateral EGFR phosphorylation (lanes 1–4 in the right panel of Figure 7F). When C29 was added 1 h after LysoPS stimulation, the patterns of ERK and EGFR phosphorylation did not change (lane 5 in the right panel of Figure 7F), indicating that TLR2 signaling is not involved in the activation of EGFR and ERK in the late phase of LysoPS-mediated signaling. In contrast, mild production of MUC5AC induced by the TLR2 agonists Pam2csk4 and Pam3csk4 was partially blocked by C29, which also moderately inhibited TLR2-dependent ERK phosphorylation (left panel in Figure 7F). These results suggest that known LysoPS receptors are dispensable for LysoPS-induced MUC5AC production. 

### 2.8. ROS Production Tends to Be Weakly Induced by LysoPS, but Is Not Involved in MUC5AC Production

As various stimuli induce MUC5AC production through ROS production [20,42,43,44], we examined whether ROS are involved in LysoPS-induced MUC5AC production. We found that LysoPS tended to induce ROS production in NCI-H292 cells but not to a statistical significance. In the presence of an ROS scavenger, NAC, but not DPI (an NADPH oxidase inhibitor), ROS levels declined by less than that was observed in unstimulated cells (Figure 8A). Varying doses of NAC or DPI failed to inhibit or alter LysoPS-induced MUC5AC production (Figure 8B). 

## 3. Discussion

LysoPS is a lysophospholipid that exerts various immunomodulatory functions primarily on immune cells, specifically mast cells. It has been postulated that this endogenous lipid molecule plays a proinflammatory role at elevated concentrations and initiates sterile inflammation in various pathologies [28,33]. In the present study, we demonstrated that LysoPS induces MUC5AC production, which is a major functional phenotype of airway epithelial cells. We investigated the mechanism of LysoPS-induced MUC5AC production with regard to signaling pathways, cognate receptors, and the oxidative stress dependency. LysoPS induces biphasic ERK activation, which is equally dependent on TACE and EGFR activities. While ERK activation in the early phase primes activation of the late phase, late-phase ERK activation directly promotes MUC5AC production. The late resurgence of ERK phosphorylation is accompanied by TGF-α and other EGFR ligand synthesis and a strong EGFR phosphorylation response. MUC5AC production is almost completely abolished by inhibitors of TACE, EGFR, and ERK. (Figure 4). We found that none of the identified LysoPS GPCR receptors is expressed in airway epithelial cells. Furthermore, a TLR2 antagonist or ROS modifiers minimally affects LysoPS-induced activation of ERK and subsequent MUC5AC production. Collectively, these results suggest that LysoPS induces MUC5AC hyperproduction in airway epithelial cells through a mechanism that involves shedding EGFR proligands and the autocrine activation of the EGFR cascade in a receptor- and ROS-independent manner. 

TACE can be activated through both external and internal routes. Phosphatidylserine (PS), which is exposed on the outer plasma membrane of apoptotic cells and is also a biosynthetic precursor of LysoPS, interacts with the membrane proximal domain of TACE to direct TACE to its target, enabling sheddase activity [45]. In the study, soluble PS inhibits the sheddase activity of TACE, presumably by impeding this interaction of PS with the TACE domain. However, in our experimental settings, soluble PS had no effect on LysoPS-dependent MUC5AC synthesis or ERK phosphorylation (data not shown), suggesting that LysoPS is unlikely to activate TACE through direct interaction with the ectodomain of TACE. Previous reports have revealed that many stimuli, notably neutrophil elastase and PMA, induce ROS, which in turn activates TACE, leading to MUC5AC production [20,42,46]. In contrast to those studies, LysoPS did not induce significant ROS production. Moreover, neither DPI nor NAC inhibited LysoPS-induced MUC5AC production, suggesting that ROS is unlikely to be involved in LysoPS-induced TACE activation. ROS-independent mucin production has also been reported in mold protease-exposed airway epithelial cells [47]. 

To date, four GPCRs, including GPR34, P2Y10, GPR174, and G2A, have been annotated as receptors of LysoPS [25,27]. LysoPS activates these receptors at a few micromolar concentrations that are not cytotoxic [38,48,49]. Moreover, the functional effects of LysoPS are often blocked by inhibitors of specific G proteins coupled with each GPCR [50]. In our hands, NCI-H292 cells do not express any of the four GPCRs, as determined by FACS and western blot analysis. Activation of TLR2 induced by LysoPS at a concentration of 1 μM is shown to be highly dependent on the chain length of the fatty acid of LysoPS [51]. LysoPS with very long chain lipid tails (C ≥ 22) induces a proinflammatory response via TLR2, whereas long chain LysoPS (C = 16–20) induces ERK phosphorylation seemingly via GPCRs. However, blocking of TLR2 using its antagonist failed to inhibit LysoPS-induced mucin production or ERK phosphorylation in our conditions (Figure 7). Amphipathic substances can activate TACE through enhanced molecular movement mediated by interactions with membrane [52,53]. Because the critical micelle concentration (cmc) of LysoPS is reported to be ~10 μM [54,55], 30 μM of LysoPS, which was used in our experiments, can promote micelle formation. In this context, cone-shaped LysoPS with an amphipathic nature can readily intercalate into membrane and influence the rate of molecular movement, including interactions between TACE and its substrate proTGF-α, initiating signaling activity of LysoPS in airway epithelial cells (Figure 9). Also, TACE can be activated intracellularly by activated ERK-dependent phosphorylation of the cytoplasmic domain of TACE [56]. In this regard, we cannot rule out the possibility that the amphipathic nature of LysoPS triggers a curvature stress in the membrane, leading to the subsequent ERK and TACE activation in a receptor independent manner. The rapid, robust kinetics of ERK phosphorylation elicited by LysoPS might also support such a mode of action of LysoPS.

As a minor lysophospholipid, LysoPS occurs at submicromolar concentrations in human and mouse plasma [28]. Currently, there is little information about the level and pathological role of LysoPS in inflamed lungs. Several enzymes that specifically synthesize and degrade LysoPS have recently been identified [50,51,57]. PS-PLA1 is a crucial enzyme that synthesizes LysoPS extracellularly [58]. It occurs at elevated levels in serum from patients with autoimmune diseases and such levels are associated with disease activity [59]. Elevated PS-PLA1, therefore, presumably increases LysoPS production in pathological conditions, where inflammatory cells and apoptotic cells, notably platelets, can provide significant amounts of LysoPS. 

Lysophospholipids function as “conditional DAMPs” when their concentrations reach pathophysiological levels [34]. One of lysophospholipids, LysoPS, provokes copious amounts of mucin in airway epithelial cells, potentially contributing to the pathophysiology of obstructive airway diseases. We identified that TACE is a key mediator that detects the proinflammatory LysoPS and communicates with EGFR via a surface signaling cascade, as evidenced by EGFR transactivation by many stimuli inducing MUC5AC production. However, LysoPS-induced MUC5AC production appears to be independent of known receptors and ROS. The nature of the initiating phase of LysoPS stimulation remains elusive. Mechanistic details of the initiating phase of LysoPS activity warrants further investigation.

## 4. Materials and Methods

### 4.1. Materials

18:1. lysophosphatidylserine (LysoPS), 18:1 lysophosphatidylcholine (LysoPC), and 18:1 lysophosphatidylethanolamine (LysoPE) were purchased from Avanti Polar Lipids (Alabaster, AL, USA). Epidermal growth factor (EGF) and TGF-α neutralizing antibody were obtained from R&D Systems (Minneapolis, MN, USA). EGFR neutralizing antibody (clone LA1), Z-VAD-fmk, BAY11-7082, diphenyleneiodonium chloride (DPI), N-acetyl-L-cysteine (NAC), TAPI2, suramin, and A438079, Ac-YVAD-cmk were purchased from Sigma-Aldrich (St. Louis, MO, USA). U0126 and AG1478 were purchased from Calbiochem (La Jolla, CA, USA). SB203580 and SP600125 were obtained from Tocris Bioscience (Bristol, UK). PTX, a Gi/o protein inhibitor, was supplied by Invitrogen Life Technologies (Carlsbad, CA, USA). C29 was purchased from Biovision (Milpitas, CA, USA).

### 4.2. Cell Cultures for NCI-H292 and Human Bronchial Epithelial Cells

The NCI-H292 cell line, human lung mucoepidermoid carcinoma, was purchased from the Korean Cell Line Bank (Seoul, Korea), and cultured in RPMI-1640 medium (Welgene, Gyeongsan, Korea) containing 10% fetal bovine serum (FBS), penicillin (100 units/mL) and HEPES (25 mM). NCI-H292 cells were cultured in a medium containing 1% FBS overnight and treated with LysoPS for the indicated time periods. Cells were pretreated with the inhibitors for 30 min prior to LysoPS treatment. Normal human bronchial epithelial (NHBE) cells (Cambrex Bio Science, Baltimore, MD, USA) were cultured in a bronchial epithelial cell growth medium (BEGM) (LONZA, Walkersville, MD, USA). The NHBE cells differentiated in medium containing a 1:1 mixture of Dulbecco’s modified Eagle medium and BEGM. After the cells attained complete confluence, the apical medium was removed for air-liquid interface culturing for 14 to 21 days, with the medium refreshed three times a week.

### 4.3. RNA Isolation, Reverse Transcription-PCR, and Quantitative Real-Time PCR 

Total mRNA was extracted from NCI-H292 and NHBE cells using a TRI reagent (Molecular Research Center, Cincinnati, OH, USA). First-strand cDNA was synthesized from 2 μg of total RNA using the LeGene express first-strand cDNA synthesis system (LeGene Biosciences, San Diego, CA, USA) in a 20 μL reaction. Reverse transcription was performed at 42 °C for 1 h, followed by heat inactivation at 70 °C for 15 min. Quantitative real-time polymerase chain reaction (PCR) amplification was performed in a 20 μL reaction with 1 μL of cDNA, 1 μL of each primer (1 μM), and 10 μL of a SYBR Green Master Mix (ELPIS Bio, Daejeon, Korea) using the QuantStudio 3 real-time PCR system (Applied Biosystems, Foster City, CA, USA). The PCR conditions were: 10 min at 95 °C, followed by 40 cycles of 95 °C for 30 s, and 60 °C for 1 min. The specificity of amplification was confirmed with a melting curve analysis. Relative expression was evaluated using the comparative cycle threshold (2-DDCt) method and expressed as the mean ± standard error of the mean (SEM). The PCR primers for MUC5AC were forward: 5′-TCCACCATATACCGCCACAGA-3′ and reverse: 5′-TGGACGGACAGTCACTGTCAAC-3′. PP1A was amplified using primers, forward: 5′-TCCTGGCATCTTGTCCATG-3′ and reverse: 5′-CCATCCAACCACTCAGTCTTG-3′.

### 4.4. TACE siRNA Transfection

TACE-specific siRNAs and TACE PCR primers were purchased from Bioneer (Daejeon, Korea). NCI-H292 cells were transfected with TACE siRNAs or control siRNA by using Lipofectamine 2000 (Invitrogen) following manufacturer’s protocols. Cells were stimulated with LysoPS at 48 h after siRNA transfection.

### 4.5. Western Blot Analysis 

Cells were lysed in a RIPA buffer (150 mM NaCl, 50 mM Tris-Cl, 0.1% NaN_3_, 1% NP-40, 0.25% sodium deoxycholate, 1 mM EDTA, 1 mM Na_3_VO_4_, 1 mM NaF and a protease inhibitor mixture). Total protein concentrations were determined using a BCA assay. The total cell lysates were resolved by SDS-PAGE and transferred to a PVDF membrane. The membrane was blocked with 5% non-fat dry milk and probed with antibodies, including anti-phospho-ERK1/2 (Cell Signaling Technology, Beverly, MA, USA), anti-phospho-EGFR (12A3: Santa Cruz Biotechnology, Dallas, TX, USA), anti-TLR2 (Santa Cruz Biotechnology), anti-ERK1/2 (Cell Signaling Technology), and anti-EGFR (1005, Santa Cruz Biotechnology). Signals from western blot analysis were visualized with an ECL detection system (Amersham Pharmacia Biotech, Piscataway, NJ, USA). 

### 4.6. Cell Viability Assays

Cell viability was determined using a CELLOMAX viability assay kit (Precaregene, Anyang, Korea). Briefly, NCI-H292 cells in 96-well plates were treated with varying concentrations of lysophospholipids for 24 h and incubated with CELLOMAX reagents for 2 h at 37 ℃.

### 4.7. Flow Cytometry

NCI-H292 cells were incubated with the indicated antibodies and analyzed using a FACSCalibur flow cytometer (BD Biosciences, San Jose, CA, USA). Data were processed by Cell-Quest software (BD Biosciences, San Jose, CA, USA). The antibodies utilized in the FACS analysis included anti-GPR34 (R&D Systems, Minneapolis, MN, USA), anti-P2Y10 (R&D Systems, Minneapolis, MN, USA), anti-GPR174 (Abcam, Cambridge, UK), and Anti-G2A (Abcam, Cambridge, UK).

### 4.8. Immunocytochemistry

Immunocytochemistry (ICC) was carried out as described previously [44]. Briefly, NCI-H292 cells were fixed in 4% paraformaldehyde, permeabilized in TBS containing 1% saponin for 30 min, and incubated at 4 °C with an anti-MUC5AC antibody (Sigma-Aldrich). After treatment of biotinylated anti-mouse IgG and then an ABC complex (Vector Laboratories, Burlingame, CA, USA), the staining was visualized using a DAB substrate kit (Thermo Fisher Scientific, Waltham, MA, USA).

### 4.9. ELISA of Secreted TGF-α and MUC5AC

NCI-H292 cells were preincubated with EGFR-neutralizing antibodies for 30 min and then stimulated with LysoPS for different periods of time. Culture supernatants were collected and the level of TGF-α was measured by a TGF-α ELISA kit (Abcam, Cambridge, UK). To measure the production of MUC5AC, NHBE cells were incubated with LysoPS for 8 h, and the culture supernatants were analyzed by a MUC5AC ELISA kit (Novus Biologicals, Littleton, CO, USA).

### 4.10. Analysis of Cellular ROS Levels

Cellular ROS were measured using the 2′,7′-dichlorofluorescin diacetate (DCFDA) cellular ROS detection assay kit (Abcam, Cambridge, UK) according to the manufacturer’s instructions. Briefly, NCI-H292 cells were plated at a density of 2 × 10^4^ cells/well into 96-well plates, stained with 20 µM of DCFDA for 45 min, and stimulated with LysoPS for 1 h. Signals were detected at Ex/Em: 485/535 mm using a Synergy H1 microplate reader (BioTek Instruments, Winooski, VT, USA)

### 4.11. Statistical Analysis 

All data were analyzed using Student’s t-tests. Differences with a *p*-value < 0.05 were considered statistically significant. Results are expressed as the mean ± SEM.

## Figures and Tables

**Figure 1 ijms-23-03866-f001:**
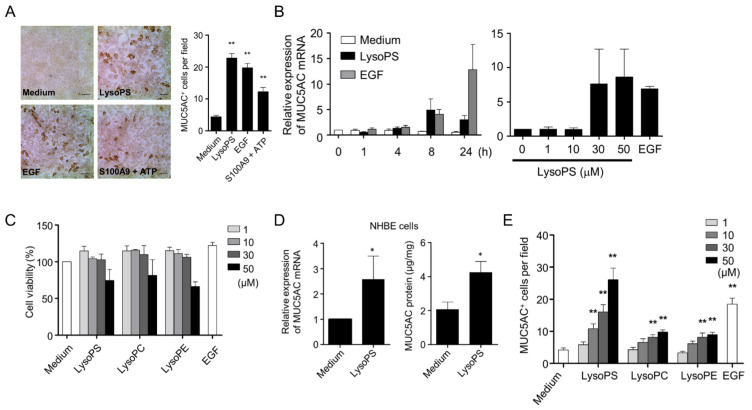
LysoPS induces MUC5AC production in airway epithelial cells. (**A**) NCI-H292 cells were treated with LysoPS (30 μM), EGF (25 ng/mL), or S100A9 (200 ng/mL) plus ATP (1 mM) for 24 h. ICC was performed for quantitative measurement of MUC5AC-positive cells. MUC5AC-positive cells were indicated as representative images (left panel), and enumerated in 9 HPFs (right panel). Data are shown as mean ± SEM of three independent experiments (** *p* < 0.01). Scale bars represent 100 μm. (**B**) *MUC5AC* mRNA were determined by real time qPCR in NCI-H292 cells treated with either LysoPS or EGF for the indicated time intervals (left panel, *n* = 4) or with different concentrations of LysoPS for 8 h (right panel, *n* = 3). The expression levels are displayed as fold increases relative to untreated cells. Data are shown as means ± SEM. (**C**) Cell viability was determined by WST-8 assays with cells treated with varying concentrations of the indicated lysophospholipids or EGF for 24 h. Cell viability was expressed as %, in which the OD value of cells cultured with medium was set to 100%. The results are shown as the mean ± SEM of three independent experiments performed in duplicates. (**D**) NHBE cells were treated with LysoPS for eight hours and analyzed for *MUC5AC* mRNA and MUC5AC protein expression. *MUC5AC* mRNA were determined by real time PCR (*n* = 4, * *p* < 0.05) and normalized by *PP1A*. MUC5AC protein levels in the supernatant were measured by ELISA (*n* = 4, * *p* < 0.05). (**E**) NCI-H292 cells were treated with varying concentrations of lysophopholipids and EGF for 24 h and then analyzed for MUC5AC-positive cells by ICC. This result is representative of three independent experiments and expressed as the mean ± SEM of MUC5AC-positive cells enumerated in 9 HPFs (* *p* < 0.05 and ** *p* < 0.01 compared with unstimulated cells).

**Figure 2 ijms-23-03866-f002:**
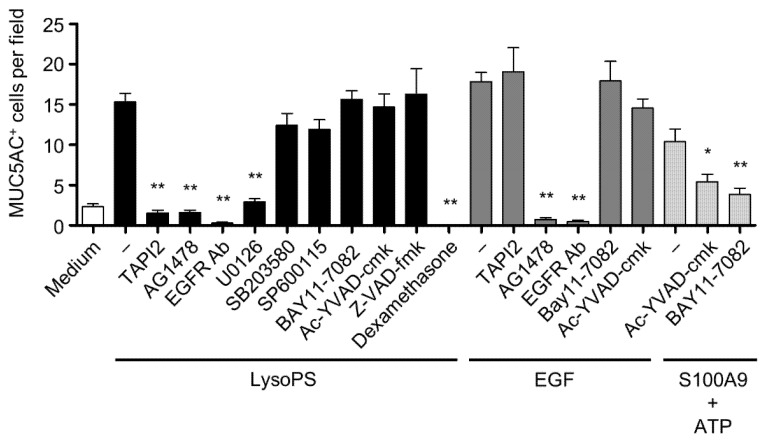
LysoPS-induced MUC5AC is abolished by inhibitors of TACE, EGFR, and ERK, but not by inhibitors of NF-κB and caspase-1. NCI-H292 cells were pretreated with inhibitors for 30 min and then treated with LysoPS, EGF or S100A9 plus ATP for 24 h for MUC5AC production determined by ICC. MUC5AC-positive cells were enumerated in 9 HPFs and shown as the mean ± SEM of 2–7 independent experiments (* *p* < 0.05, ** *p* < 0.01). The following inhibitors were used: TAPI2 (10 μM), AG1478 (10 μM), U0126 (10 μM), SB203580 (10 μM), SP600125 (20 μM), Ac-YVAD-cmk (10 μM), Z-VAD-fmk (10 μM), BAY11-7082 (10 μM), dexamethasone (1 μg/mL), and EGFR neutralizing antibody (10 μg/mL).

**Figure 3 ijms-23-03866-f003:**
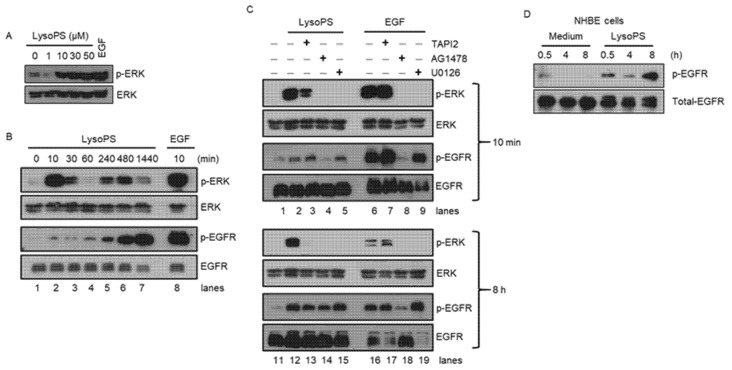
LysoPS induces a biphasic ERK phosphorylation that is blocked by inhibitors of TACE, EGFR, and ERK pathways. (**A**) NCI-H292 cells were treated with different concentrations of LysoPS for 10 min and subjected to immunoblot analysis against ERK phosphorylation. This blot is a representative of two independent experiments. (**B**) NCI-H292 cells were treated with LysoPS for different time intervals and examined for ERK and EGFR phosphorylation. The cells treated with EGF for 10 min were used as a positive control. This blot is a representative of three independent experiments. (**C**) NCI-H292 cells were pretreated with TAPI2, AG1478, or U0126 for 30 min and then treated with LysoPS or EGF for 10 min or 8 h. (**D**) NHBE cells were stimulated by LysoPS for the indicated time points and the levels of EGFR phosphorylation were examined. This result is a representative of three independent experiments. Each blot was reprobed with anti-ERK and/or anti-EGFR for loading controls.

**Figure 4 ijms-23-03866-f004:**
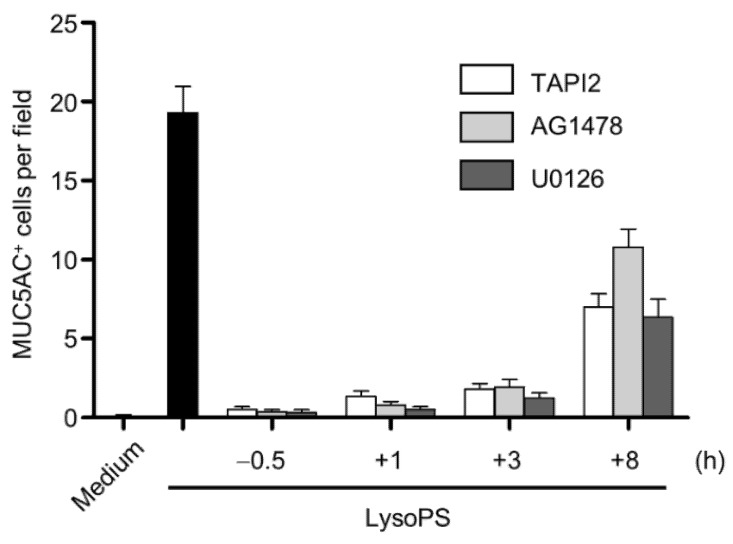
The TACE-EGFR-ERK pathway at the late phase is required for MUC5AC production. The indicated inhibitors were added to NCI-H292 cells at either 30 min before or one, three, and eight h after treatment with LysoPS. MUC5AC-positive cells were evaluated 24 h after treatment with LysoPS. MUC5AC positive cells were enumerated in 8–9 HPFs and expressed as the mean ± SEM of two independent experiments.

**Figure 5 ijms-23-03866-f005:**
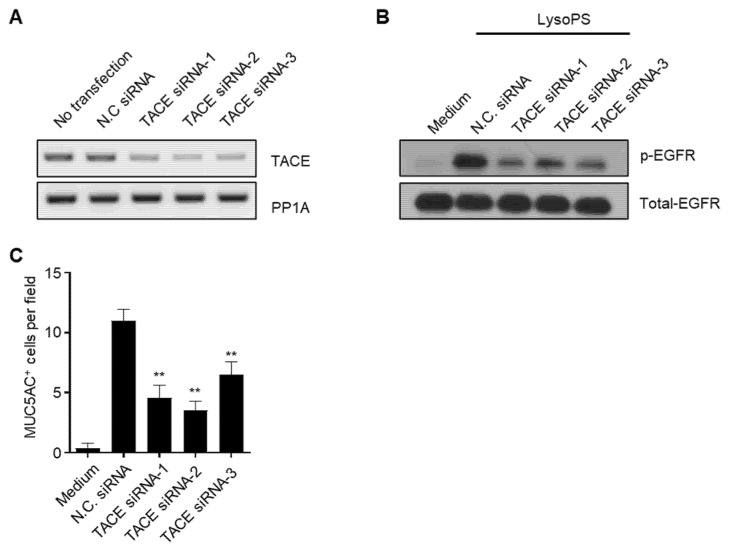
Knockdown of TACE gene expression reduces LysoPS-dependent EGFR phosphorylation and MUC5AC production. NCI-H292 cells were treated with siRNAs (40 nM) or Control siRNA (NC, 40 nM) for 48 h. (**A**) RT-PCR was performed to validate the effects of siRNAs against TACE mRNA. PP1A was used as a loading control. (**B**) Cells were treated with LysoPS for 8 h and examined for EGFR phosphorylation. The blot was reprobed with anti-EGFR for a loading control. (**C**) 24 h after LysoPS treatment, MUC5AC production determined by ICC. MUC5AC-positive cells were enumerated in 9 HPFs and expressed as the mean ± SEM of two independent experiments (** *p* < 0.01).

**Figure 6 ijms-23-03866-f006:**
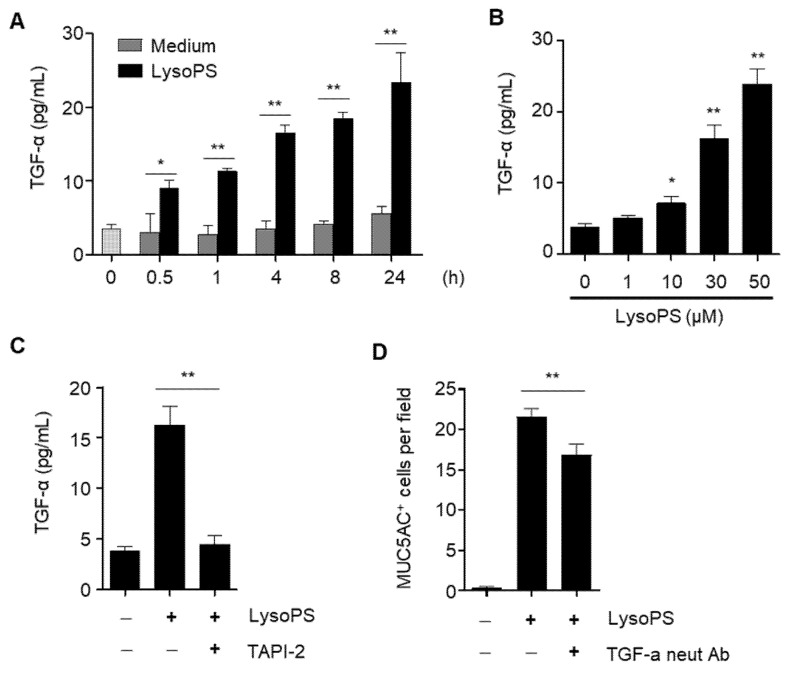
LysoPS induces TGF-α secretion in a TACE-dependent manner. NCI-H292 cells were treated with LysoPS (30 μM) for different time intervals (**A**) or with different concentrations of LysoPS for 8 h (**B**) or pretreated with TAPI2 for 30 min and then stimulated with LysoPS for 8 h (**C**). TGF-α in the culture supernatant was measured by ELISA. *n* = 2–5 for A, *n* = 5 for B and C, * *p* < 0.05, ** *p* < 0.01. (**D**) NCI-H292 cells were pretreated with TGF-α neutralizing antibody (2 μg/mL) for 30 min before determining LysoPS- triggered MUC5AC producing cells. MUC5AC-positive cells were enumerated in 9 HPFs and expressed as the mean ± SEM of three independent experiments.

**Figure 7 ijms-23-03866-f007:**
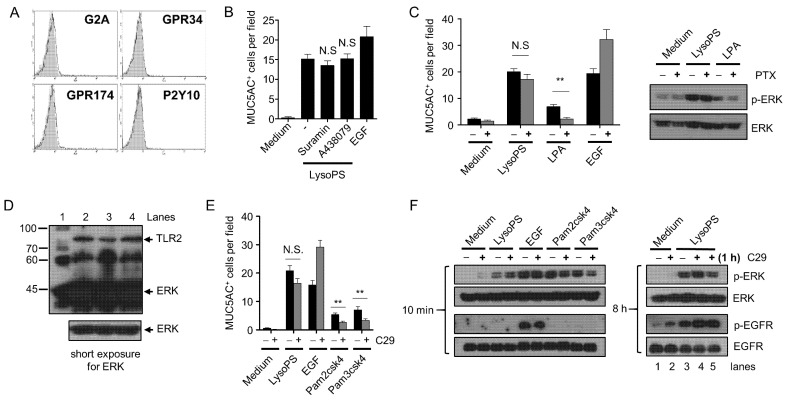
None of the known receptors for LysoPS is engaged in LysoPS-induced MUC5AC production. (**A**) The expression of G2A, GPR34, GPR174, and P2Y10 on the surface of NCI-H292 cells ware analyzed by FACS. Staining with specific antibodies (solid lines) and isotype control antibodies (gray) are shown. (**B**) NCI-H292 cells were pretreated with suramin (10 μM) or A438079 (10 μM) for 30 min and then stimulated for 24 h with LysoPS or EGF. MUC5AC-positive cells were enumerated in nine HPFs and expressed as the mean ± SEM of two independent experiments. (**C**) NCI-H292 cells were treated for 10 min (for immunoblot analysis) or 24 h (for ICC) with LysoPS, LPA (30 μM), or EGF in the absence or presence of PTX (100 ng/mL). MUC5AC positive cells were enumerated in 9 HPFs and expressed as the mean ± SEM of two to three independent experiments (** *p* < 0.01 and NS = not significant). ERK phosphorylation and total ERK was analyzed by immunoblot analysis. (**D**) Total cell lysate of NCI-H292 cells that had been treated with LysoPS or EGF for 10 min were subjected to immunoblot analysis against TLR2 and ERK. Lane 1 for size marker, lane 2 for medium, lane 3 for LysoPS, and lane 4 for EGF. (**E**,**F**) NCI-H292 cells were treated with LysoPS, EGF, Pam2csk4 (1 μg/mL), or Pam3csk4 (1 μg/mL) in the absence or presence of C29 (40 μM) for 24 h (for ICC) or for 10 min and 8 h (for immunoblot analysis). MUC5AC-positive cells were enumerated in 9 HPFs and expressed as the mean ± SEM of two independent experiments (** *p* < 0.01 and NS = not significant). ERK phosphorylation and total ERK were analyzed by immunoblot analysis. Lanes 1–4 in *F* indicate that NCI-H292 cells were treated LysoPS for 8 h in the absence or presence of C29 (40 μM), while lane 5 indicates that C29 was added to the culture 1 h after treatment with LysoPS and incubated for an additional 7 h. These results are representative of three independent experiments.

**Figure 8 ijms-23-03866-f008:**
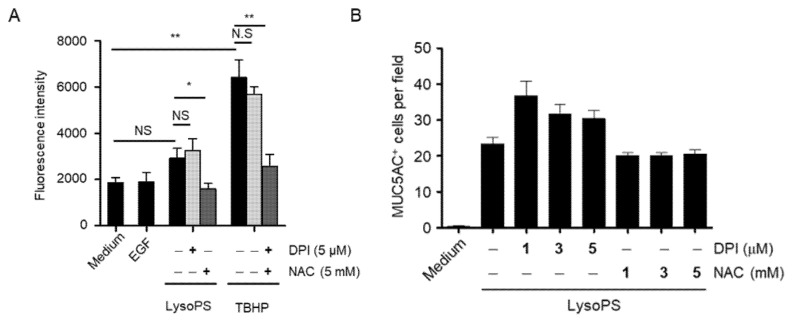
ROS is dispensable for LysoPS-induced MUC5AC production (**A**) In the absence or presence of DPI (5 μM) or NAC (5 mM), NCI-H292 cells were treated for 1h with EGF or stimulated with LysoPS and TBHP (100 μM). ROS production was measured by DCFDA fluorescence intensity (*n* = 2–4, * *p* < 0.05, ** *p* < 0.01, NS = not significant). (**B**) NCI-H292 cells were treated for 24 h with LysoPS in the absence or presence of DPI or NAC. MUC5AC-positive cells were enumerated in 9 HPFs and expressed as the mean ± SEM of two independent experiments.

**Figure 9 ijms-23-03866-f009:**
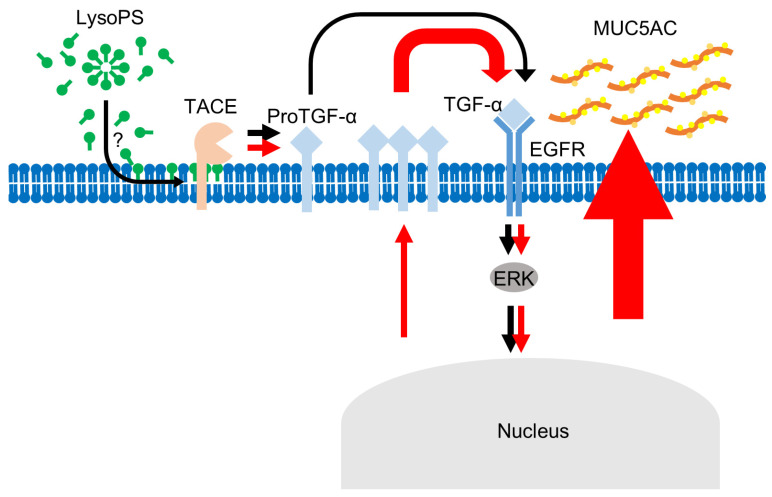
Schematic of LysoPS-induced MUC5AC production. In the early phase (black arrows), LysoPS activates the membrane-associated TACE, leading to maturation of a pre-existing small pool of pro-TGF-α. This process appears to be receptor-independent but dependent on the amphipathic nature of LysoPS that causes membrane perturbation. The resulting mature TGF-α signals through EGFR, which activates ERK in the early phase, followed by the induction of new TGF-α. In the late phase (red arrows), a large amount of newly synthesized TGF-α promotes the prolonged activation of ERK via EGFR and drives ERK-dependent MUC5AC production in airway epithelial cells.

## Data Availability

Data is contained within the article or Appendix A.

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
