# Peer review of "Lysophosphatidylserine Induces MUC5AC Production via the Feedforward Regulation of the TACE-EGFR-ERK Pathway in Airway Epithelial Cells in a Receptor-Independent Manner"

_ijms, 2022, doi:10.3390/ijms23073866_

Round 1
Reviewer 1 Report
In this study, the authors present a new proinflammatory function of LysoPS as a potent inducer of MUC5AC production in airway epithelial cells, this function appears to be mediated by positive feedback regulation of the TACE-EGFR-ERK pathway. The study is complex and structured with well-conducted experiments, nevertheless it has some weaknesses, that need to be improved.
The abstract and the introduction in their present form are not completely clear. It should be stated from the beginning what was the purpose of this study. In particular:
- The introduction is too long, and some parts are unnecessary, this loses the aim of the study that should be described more clearly at the end of the introduction.
- The same thing is true for abstract that must go directly to the point as a description of aim, methods and results.
- The authors should implement the description of the statistical analysis that are very poor.
- Many corrections have to be made in the language, punctuation and syntax of the text.
Reviewer 2 Report
In this study, the authors evaluated the mechanisms responsible for hypersecretion of mucins in airway epithelial cells (AEC). For this, they measure the effect of stimulation by lysophosphatidylserine (lysoPS). The subject is original and the data are new and convincing. However, the authors did not justify their approach.
Major comments
- As mentioned above, the authors did not explain the reason for chosing lysoPS as a stimulus in the introduction.
- The authors used the NCI-H292 cells, a mucoepidermoid carcinoma, for their experiments with some validation with NHBE. However, they did not show that lysoPS induced MUC5AC production By EGFR activation. Please, the authors should control this.
- LysoPS seems to be effective at 30-50 µM. Is this a concentration usually found in lung tissue? Moreover, this compound induces some cytotoxic effect at 50µM concentration. We can suspect that these concentrations could provok some cell stress responsible for the EGFR activation?
- Activation of TACE should be measured by enzymatic activity. The authors should check the enzymatic activity. Is it associated with the release of TNF-alpha as an example. The authors should try to inhibit TACE by another method such as siRNA in order to verify that TACE is involved in EGFR activation.
- The implication of TGF-alpha in EGFR activation should be verified by another method (either the use of neutralizing antibody or specific siRNA).
- Since lysoPS is exposed on cell membrane of apoptotic cells, does the contact with apoptotic cells induces the secretion of MUC5AC? Alternatively, does induction of apoptosis in AEC induces the hypersecretion of mucins in valid neighbourhood cells?
Minor comments
1. Please, specify if the method of measurement of MUC5AC protein is really quantitative? A dot blot method night also be used.
